# Amino Phenyl Copper Phosphate-Bridged Reactive Phosphaphenanthrene to Intensify Fire Safety of Epoxy Resins

**DOI:** 10.3390/molecules28020623

**Published:** 2023-01-07

**Authors:** Huiyu Chai, Weixi Li, Shengbing Wan, Zheng Liu, Yafen Zhang, Yunlong Zhang, Junhao Zhang, Qinghong Kong

**Affiliations:** 1School of Emergency Management, Jiangsu University, Zhenjiang 212013, China; 2Zhejiang Jiamin New Materials Co., Ltd., Jiaxing 314027, China; 3School of Environmental and Chemical Engineering, Jiangsu University of Science and Technology, Zhenjiang 212003, China

**Keywords:** amino phenyl metal phosphate, organic–inorganic hybrid, reactive flame retardant, epoxy resin

## Abstract

To improve the compatibility between flame retardant and epoxy resin (EP) matrix, amino phenyl copper phosphate-9, 10-dihydro-9-oxygen-10-phospha-phenanthrene-10-oxide (CuPPA-DOPO) is synthesized through surface grafting, which is blended with EP matrix to prepare EP/CuPPA-DOPO composites. The amorphous structure of CuPPA-DOPO is characterized by X-ray diffraction and Fourier-transform infrared spectroscopy. Scanning electron microscope (SEM) images indicate that the agglomeration of hybrids is improved, resisting the intense intermolecular attractions on account of the acting force between CuPPA and DOPO. The results of thermal analysis show that CuPPA-DOPO can promote the premature decomposition of EP and increase the residual amount of EP composites. It is worth mentioning that EP/6 wt% CuPPA-DOPO composites reach UL-94 V-1 level and limiting oxygen index (LOI) of 32.6%. Meanwhile, their peak heat release rate (PHRR), peak smoke production release (PSPR) and CO_2_ production (CO_2_P) are decreased by 52.5%, 26.1% and 41.4%, respectively, compared with those of EP. The inhibition effect of CuPPA-DOPO on the combustion of EP may be due to the release of phosphorus and ammonia free radicals, as well as the catalytic charring ability of metal oxides and phosphorus phases.

## 1. Introduction

Epoxy resin (EP) has been widely used in many fields because of its excellent performance [1,2]. Nevertheless, its highly flammable and asphyxiating fumes cannot meet the needs of sophisticated fields [3,4]. Therefore, improving the fire resistance of epoxy resin is an important goal in the field of intrinsic safety [5,6].

In recent years, two-dimensional layered metallic phosphates have attracted extensive attention owing to their regular layered structures and excellent flame-retardant properties [7,8]. Phenyl metal phosphonates, as a class of layered metallic phosphates, have been a research focus in the field of flame-retardant materials because of their stable chemical properties, suitable thermal stability and good compatibility with matrix [9,10]. For example, UL-94 of V-0 grade was observed with 1 wt% layered lanthanum phenyl phosphate (LaHPP) and 7 wt% decabromodiphenyl oxide (DBDPO) in a polycarbonate formulation, which was attributed to the lamellar structure and condensation phase reinforcement of lanthanum phenyl phosphate [11]. It was reported that the flue gas and toxic gas release of EP composites with the addition of copper phenyl phosphate (CuPP) was reduced [12]. However, phenyl metal phosphonates are difficult to uniformly disperse in polymer matrix due to their unique laminated structure, which has a great influence on the flame retardancy of polymer composites [13,14]. Therefore, it is necessary to improve the dispersion of phenyl metal phosphonate in polymer matrices.

Reactive flame retardants can effectively solve the problem of poor dispersion in polymer composites [15]. DOPO, as a reactive flame retardant, has good oxidation resistance and easily modifiable property [16]. The inherent P-H bond in the structure of DOPO has high reactivity, and so it can be reacted with unsaturated bonds and epoxy groups to form various derivatives, which can be cross-linked with EP matrix [17]. Furthermore, nitrogenous DOPO compounds synthesized by the addition reaction of DOPO and imine compounds are the most common solidified DOPO flame retardants [18]. The advantages of DOPO-imine flame retardants lie in their simple synthesis conditions, adjustable reaction process, high product yield and molecular structure containing both phosphorus and nitrogen flame-retardant elements [19,20]. The application of DOPO combined with inorganic nanomaterials through the amino group in flame-retardant EP has been reported. The PHRR and total heat release (THR) of EP/7 wt% zirconium hybrid polyhedron oligomeric polysiloxane derivative composites decreased by 37.1% and 16.7%, respectively, compared with those of EP [21]. A piperazinyl DOPO derivative was developed with the aim of functionalizing graphene oxide to form a hybrid, which reduced the mechanical cracking of epoxy resins and improved their biphasic compatibility [22]. Therefore, reactive DOPO-imine derivatives as flame retardants for EP are worthy of further investigation.

Given this context, it can be said that reactive DOPO-imine derivatives can effectively alleviate the compatibility disadvantage of phenyl metal phosphonate, and can further improve the flame-retardance efficiency. Hence, in this work, an amino phenyl copper phosphate (CuPPA) was successfully synthesized. Then, CuPPA was further grafted with DOPO to form complex compounds. Inorganic nanoparticles were combined with an organic high-efficiency phosphorus flame retardant through this molecular structural design. Meanwhile, reactive flame-retardant technology was introduced to promote the formation of uniform and stable EP composites due to the presence of imines. The results indicated that the uniform EP composites exhibited remarkable flame retardancy when CuPPA-DOPO was introduced into the epoxy resin combustion system.

## 2. Results and Discussion

### 2.1. Structural Analysis of CuPPA-DOPO

Figure 1a shows the XRD patterns of CuPPA and CuPPA-DOPO. The diffuse scattering peak of CuPPA occurs at about 26°, indicating that its degree of crystallinity is not high and revealing its amorphous structure [23]. The diffuse scattering peak decreased slightly and the crystallinity did not change significantly after grafting DOPO onto CuPPA. FTIR spectra (Figure 1b) reveal the structures of CuPPA and CuPPA-DOPO. The absorption peaks at 1148 cm^−1^ and 1054 cm^−1^ are the stretching vibrations of P=O and P-O bonds, and the peaks at 615 cm^−1^ and 533 cm^−1^ are the characteristic peaks of C-P bonds [12]. The absorption peak at 1092 cm^−1^ corresponds to the stretching vibrations of Cu-O-P bonds [24]. The absorption peaks at 1501 cm^−1^ and 834 cm^−1^ correspond to the stretching vibrations of C=C and C-N-C bonds. The absorption peaks at 3330 cm^−1^ and 3445 cm^−1^ belong to the stretching vibrations of -NH_2_ bonds, which are consistent with the molecular formula of CuPPA [25,26]. The vibrational band occurring at 758 cm^−1^ is attributable to the absorption of P-O-Ph, and the vibrational band of P-H in DOPO disappears. The double peaks of -NH_2_ in CuPPA shift to a single peak (3335 cm^−1^) of -NH in CuPPA-DOPO [27]. The above analyses prove that DOPO was successfully introduced into the molecular chain of CuPPA. The microstructures of CuPPA and CuPPA-DOPO can be directly observed in the SEM images shown in Figure 1c,d. A rod-like nanostructure with tiny nanoparticles was clearly visible after grafting DOPO onto CuPPA, which was different from the stacked lamellar structure of CuPPA. The nanorods had a diameter of about 500 nm and a length of a few microns.

The thermal decomposition behavior of CuPPA, DOPO and CuPPA-DOPO was obtained by TGA, as plotted in Figure 2. The decomposition of CuPPA-DOPO has three stages of weightlessness. The first weight loss (5.8 wt%) occurred at 30–105 °C, and was mainly the desorption of adsorbed water and a small amount of organic solvents [28]. The next weight loss was 14.2 wt% after 105–306 °C, accompanied by the decomposition of phosphaphenanthrene into small molecules [29]. The most intense weight loss was 52.0 wt% after 306 °C, attributable to the decomposition of metal compounds and the oxidation of organic molecules such as phosphonates, the phase of which coincided with the decomposition temperature of EP [30].

### 2.2. Thermal Properties of EP Composites

The thermal stability of EP nanocomposites is an important evaluation index of fire safety. TGA and DTG curves and relevant data of EP nanocomposites are shown in Figure 3a,b and Table 1. The initial decomposition temperature (*T*_5%_) of EP was 360 °C, while EP/CuPPA-DOPO composites displayed lower *T*_5%_. Among EP composites, *T*_5%_ and *T*_max_ (the temperature at maximum pyrolysis rate) of EP/6 wt% CuPPA-DOPO composites were decreased to 333 °C and 368 °C, respectively, demonstrating that CuPPA-DOPO promoted the earlier decomposition of EP composites [31]. However, the promoting effect of CuPPA was not obvious, with *T*_5%_ and *T*_max_ of 350 °C and 376 °C. The decrease of the maximum decomposition rate of EP/6 wt% CuPPA-DOPO composites also indicates that the presence of CuPPA-DOPO delayed the degradation of the EP at a higher temperature [32]. Moreover, the incorporation of CuPPA-DOPO improved the char-forming ability of the EP composites. The residues of EP composites were increased to 17.4%, 23.0%, 24.9% and 26.1%, respectively, when the amounts of CuPPA-DOPO were 2, 4, 6 and 8 wt%. The residue of EP/4 wt% CuPPA composites at 700 °C was 19.8%, which is higher than that (14.9%) of pure EP, but lower than that of EP/4 wt% CuPPA-DOPO. The outstanding char-forming ability of EP is attributed to the polyphosphoric substances and metallic oxide produced in the early decomposition of CuPPA-DOPO [33,34].

### 2.3. Flame Retardancy of EP Composites

The results of UL-94 vertical burning and LOI test of EP/CuPPA-DOPO composites are shown in Table 2. EP composites all reached a UL-94 V-1 rating, and the LOI values increased to 28.8%, 30.8%, 32.6% and 31.4% when adding 2, 4, 6 and 8 wt% CuPPA-DOPO, respectively, while EP was evaluated as no grade and the LOI value increased only to 25.9%. EP/6 wt% CuPPA-DOPO composites had the best flame-retardant effect. The flame retardancy of EP/8 wt% CuPPA-DOPO composites was slightly poor, which may have been due to the poor dispersibility caused by the excessive dosage, degrading the fluidity of EP/CuPPA-DOPO composites and greatly promoting the combustion of EP composites [35,36]. However, UL-94 reached grade V-1 with 45.2 s, and the LOI value was only 28.2% when 4 wt% CuPPA was added, indicating that the flame retardancy of CuPPA was not as good as that of CuPPA-DOPO. In addition, some of the following features can be seen from the LOI char image in Figure 4. The residue of EP/4 wt% CuPPA-DOPO composites was dense and not easily broken, and more than that of EP/4 wt% CuPPA composites, while EP had less residue. In the system of EP/CuPPA-DOPO composites, the phosphoric acid and metal oxides generated by decomposition could also promote the formation of char layer, insulate the heat transfer, limit oxygen exchange, and thus inhibit the further combustion of EP [37,38].

In order to more intuitively and accurately reflect the combustion behavior of EP composites, CCT was conducted to obtain the heat release rate (HRR), smoke production rate (SPR), total smoke release (TSR) and CO_2_ production (CO_2_P), as displayed in Figure 5a–d and Table 3. The PHRRs of EP composites declined to 739, 617, 390 and 597 kW/m^2^ with the addition of 2, 4, 6 and 8 wt% CuPPA-DOPO, respectively, representing decreases of 10.1%, 24.9%, 52.5% and 27.4% compared with that (822 kW/m^2^) of pure EP. The PHRR of EP/4 wt% CuPPA composites was 784 kW/m^2^, 4.6% lower than that of pure EP, which indicates that CuPPA-DOPO had a better inhibitory effect on the combustion of EP than CuPPA. The decrease of PHRR can be interpreted as being a result of H_2_O, CO_2_ and NH_2_ released by the decomposition of CuPPA-DOPO diluting the concentration of combustible volatile matter in the gas phase and absorbing part of the combustion heat [39,40].

Notably, the PSPR of pure EP was 0.23 m^2^/s, and that of EP/4 wt% CuPPA composites was 0.21 m^2^/s. The PSPR of EP composites reduced to 0.21, 0.20, 0.17 and 0.20 m^2^/s when adding 2, 4, 6 and 8 wt% CuPPA-DOPO, respectively, representing decreases of 8.7%, 13.0%, 26.1% and 13.0% compared with pure EP. Meanwhile, the TSR of EP composites decreased by 28.3%, 20.3%, 13.4% and 14.5% with the addition of 2, 4, 6 and 8 wt% CuPPA-DOPO, respectively, compared with that (2438 m^2^/m^2^) of pure EP. The TSR of EP/4 wt% CuPPA composites was 1585 m^2^/m^2^, 35% lower than pure EP and 18.4% lower than EP/4 wt% CuPPA-DOPO, which is because small phospho-oxygen molecules released after CuPPA-DOPO combustion became parts of the flue gas and entered the flame, thus increasing the TSR value and degenerating the smoke suppression performance [41,42,43].

High levels of carbon dioxide in fires are a major cause of asphyxia and poisoning. The CO_2_ production (CO_2_P) curves of EP composites show that upon adding 2, 4, 6 and 8 wt% CuPPA-DOPO, the CO_2_P of EP composites decreases to 0.79, 0.70, 0.51 and 0.75 g/s, respectively, representing decreases of 9.2%, 19.5%, 41.4% and 13.8% compared with that (0.87 g/s) of pure EP. However, the CO_2_P of EP/4 wt% CuPPA composites was 0.85 g/s, slightly lower than that of pure EP and higher than that of EP/4 wt% CuPPA-DOPO. The main reason for the reduction of CO_2_P is that the combustion of CuPPA-DOPO can produce CuO and phosphorous flame retardant, catalyzing the formation of a dense carbon layer to protect the surface of the composite materials. The PO• produced by CuPPA-DOPO combustion exerts flame retardancy in the gas phase and inhibits the complete combustion of the gas-phase products [44,45].

### 2.4. Flame-Retardance Mechanism

The microstructures of the outer (Figure 6a–c) and inner (Figure 6d–f) char layers of EP composites were characterized by SEM to obtain more detailed information on the flame-retardance mechanism. The outer surface of the char layer of pure EP was obviously lumpy, with large and dense cracks, which could not play a barrier role in combustion. In addition, there were many large pores and a large number of cracks on the inner surface of the char layer of pure EP, which were caused by intense burning, with a large amount of heat and smoke escaping. More dense and hard expanded char appeared on the outer surface of EP/4 wt% CuPPA-DOPO composites’ char layer. Its inner surface char layer was thicker and rougher, with bonded particles compared with EP/4 wt% CuPPA composites. On the one hand, phosphorous groups contributed to the formation of char. On the other hand, the cured cross-linked network structure enhanced the strength of the residual coke, which could play the role of insulation barrier, reducing the transfer of heat, mass and oxygen between internal and external of EP composites [46,47].

The flame-retardance mechanism of EP/CuPPA-DOPO composites was inferred by analyzing the above SEM results of the EP composites, as shown in Figure 7. In the condensed phase, the exposed copper-metal centers can catalyze organic reactions such as dehydrogenation and esterification [48,49]. In addition, phosphoric acid derivatives also promote the formation of skeleton-stable polycyclic aromatic hydrocarbons, thereby reducing heat and mass transfer and further inhibiting combustion [50,51,52]. In the gas phase, the HPO• and PO• radicals generated during the decomposition of CuPPA-DOPO interrupt the chain reaction [53]. Meanwhile, the non-flammable gases dilute the combustible gases on the surface of EP composites, which is conducive to retarding flame spread [54,55].

## 3. Materials and Methods

### 3.1. Materials

Para-phenylene diamine (C_6_H_8_N_2_), potassium iodide (KI) and 2-chloroethyl phosphate (C_2_H_6_ClO_3_P) were supplied by Macklin Co., Ltd. (Shanghai, China). Additionally, 9,10-dihydro-9-oxygen-10-phospha-phenanthrene-10-oxide (DOPO) (phosphorus content, ≥14% by weight), copper chloride dihydrate (CuCl_2_·2H_2_O), paraformaldehyde (HCHO), tetrahydrofuran (C_4_H_8_O), sodium hydroxide (NaOH) and diaminodiphenylmethane (DDM) (viscosity at 25 °C, 2.5–4.0 Pa∙s, amine value, 480 mg KOH g^−1^) were acquired from Sinopharm Chemical Reagent Co., Ltd. (Shanghai, China). EP (NFEL128) (viscosity, 12–15 Pa∙s, epoxy equivalent, 184–190 g/mol) was bought from Nanya Electronic Materials (Kunshan) Co., Ltd. (Kunshan, China).

### 3.2. Preparation of Amino Phenyl Copper Phosphate

The schematic diagram of CuPPA synthesis is shown in Figure 8a. First, 400 mL deionized water, 7.3 g 2-chloroethyl phosphoric acid, 5.4 g p-phenylenediamine and 0.8 g KI were added into a 500 mL beaker and stirred until the mixture was completely dissolved. Next, 40 mL cold NaOH (1 mol/L) solution was dropped into a beaker and stirred for 30 min. After reacting at 18 °C for 48 h, 6.8 g CuCl_2_·2H_2_O was added into the mixture and the mixed system was continuously stirred for 4 h at 70 °C under magnetic agitation. Finally, the flask was aged for 24 h. The precipitates were filtered and washed with deionized water, then followed by vacuum drying at 80 °C for 24 h.

### 3.3. Preparation of DOPO-Amino Phenyl Copper Phosphate

The reaction process of CuPPA-DOPO is illustrated in Figure 8b. First, 100 mL tetrahydrofuran and 5 g CuPPA were added into a 250 mL three-necked flask and stirred for 0.5 h. Next, 2.5 g DOPO and 0.5 g paraformaldehyde were added into the mixture and stirred vigorously at 50 °C for 10 h, keeping the condensation reflux under nitrogen atmosphere. The products were washed three times and dried under vacuum at 80 °C for 24 h.

### 3.4. Preparation of EP/CuPPA-DOPO Composites

Firstly, CuPPA-DOPO was dispersed in acetone and sonicated for 30 min. The preheated EP matrix was added in the mixture for 30 min of ultrasonic. Next, the mixture was stirred at 90 °C for 4 h until the acetone was completely removed. Then, DDM (mass ratio of EP matrix to DDM was 4:1) was added to the dispersion and stirred until DDM was completely dissolved. Then, the mixture was quickly poured into the preheated silicone rubber mold and cured at 100 °C for 2 h. Finally, the splines were transferred to the drying oven and cured at 110 °C/2 h, 130 °C/2 h, 150 °C/2 h. The specific contents and curing reaction of EP composites are shown in Table 4 and Figure 9.

### 3.5. Measurements

X-ray diffraction (XRD) patterns of CuPPA-DOPO and CuPPA were measured with a MAX-Rb instrument produced by Rigaku Co., Ltd. (Matsumoto, Japan), which had a scanning range of 5° to 80°, a scanning speed of 5°/min and a step length of 0.02°. Fourier-transform infrared (FTIR) spectra were obtained with a 6700 spectrometer produced by Nicolet Instruments Co., Ltd. (Madison, WI, USA). The scanning electron microscopy (SEM) instrument was a JSM-7001F microscope produced by JEOL (Tokyo, Japan), and gold spraying was carried out before testing. Vertical burning (UL-94) tests were performed with a CZF-1 vertical combustion instrument produced by Jiang Ning Analytical Instrument Co., Ltd. (Nanjing, China). The spline size was 130 × 13 × 3.2 mm^3^, and the test standard was in accordance with ASTM D 3801. Limiting oxygen index (LOI) was tested by a JF-3 oxygen index instrument produced by Jiang Ning Analytical Instrument Co., Ltd. (Nanjing, China). The spline size was 130 × 6.5 × 3.2 mm^3^ and the test standard was in accordance with ASTM D 2863. Thermogravimetric analysis (TGA) was conducted using a thermogravimetric analyzer produced by Mettler Toledo, with a sample quantity of about 10 mg at a ramp rate of 10 °C/min in nitrogen environment. Burning data of samples were obtained using the cone calorimeter test (CCT) with sheet dimensions of 100 × 100 × 3.0 mm^3^ according to ISO 5660-1. The spline was wrapped in aluminum foil and placed at a horizontal heat flux of 35 kW/m^2^.

## 4. Conclusions

In summary, DOPO-amino phenyl copper phosphate was successfully synthesized, which was incorporated into EP matrix for preparing uniformly dispersed EP/CuPPA-DOPO composites. FTIR and SEM characterizations showed that CuPPA-DOPO was an amorphous rod-like nanoparticle. TGA results showed that the residue of EP/8 wt% CuPPA-DOPO composites was increased to 26.1%, about 11.2% higher than that of pure EP. Moreover, the obtained EP nanocomposites exhibited noteworthy flame-retardant properties with low additives. EP composites reached a UL-94 V-1 rating with a dosage of CuPPA-DOPO as low as 2 wt%. The PHRR, PSPR, TSR and CO_2_P of EP/CuPPA-DOPO were all decreased compared with those of EP. The inhibition of CuPPA-DOPO on EP combustion was attributed to a char barrier in the condensed phase, quenching by releasing some phosphorus molecular debris in the gas phase as well as the synergy of phosphorous and nitrogen to dilute combustible gases. Therefore, the multi-synergistic flame-retardance mechanism of CuPPA-DOPO is of great significance in the preparation of organic and inorganic hybrid flame retardants.

## Figures and Tables

**Figure 1 molecules-28-00623-f001:**
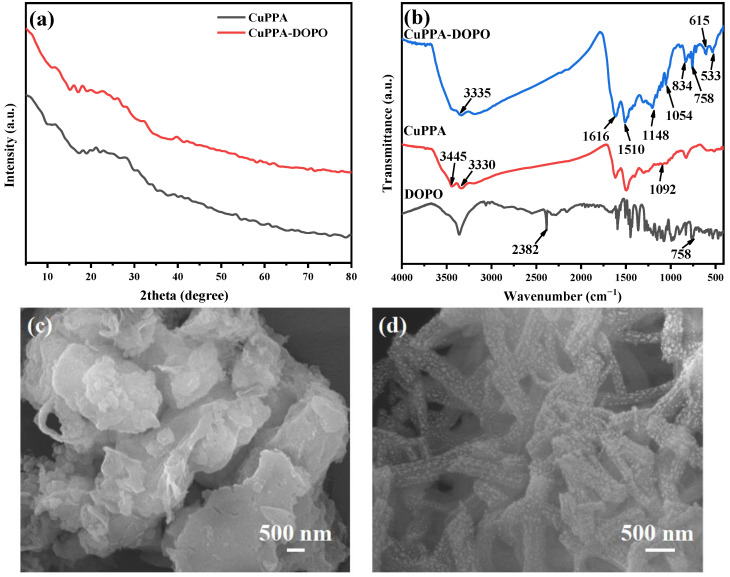
(**a**) XRD patterns of CuPPA and CuPPA-DOPO; (**b**) FTIR spectra of CuPPA and CuPPA-DOPO; (**c**) SEM image of CuPPA; (**d**) SEM image of CuPPA-DOPO.

**Figure 2 molecules-28-00623-f002:**
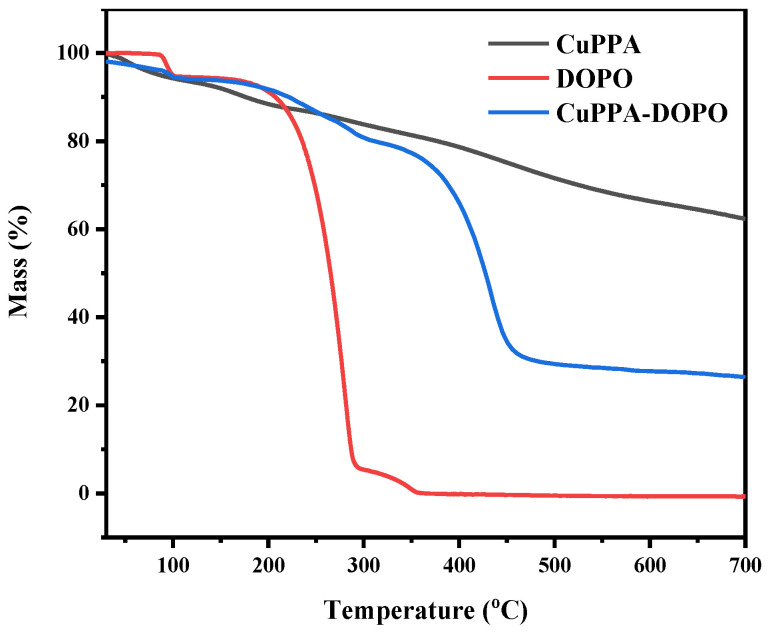
TGA curves of CuPPA, DOPO and CuPPA-DOPO.

**Figure 3 molecules-28-00623-f003:**
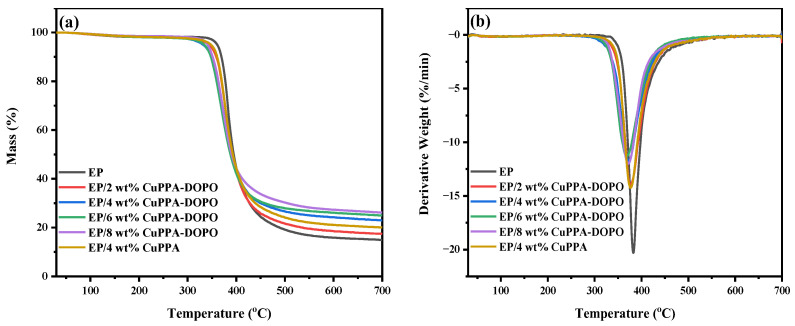
(**a**) TGA and (**b**) DTG curves of pure EP and EP composites.

**Figure 4 molecules-28-00623-f004:**
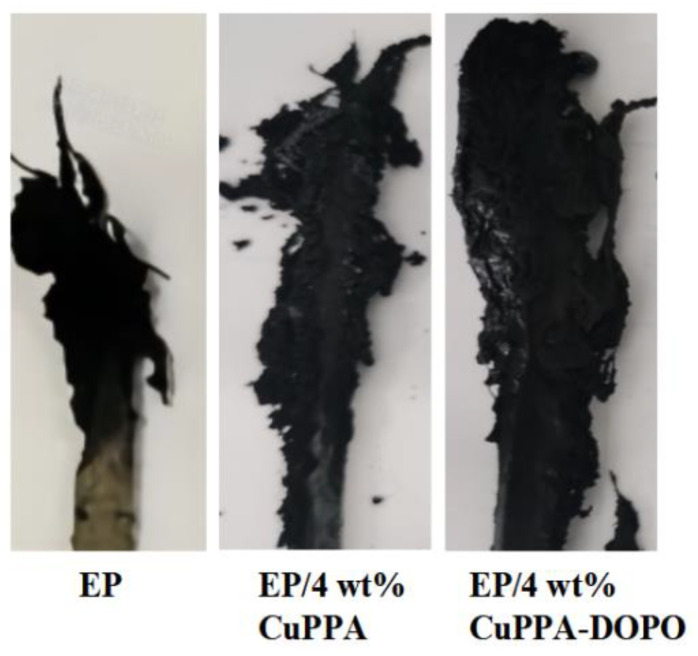
Photographs of pure EP and EP composites after LOI test.

**Figure 5 molecules-28-00623-f005:**
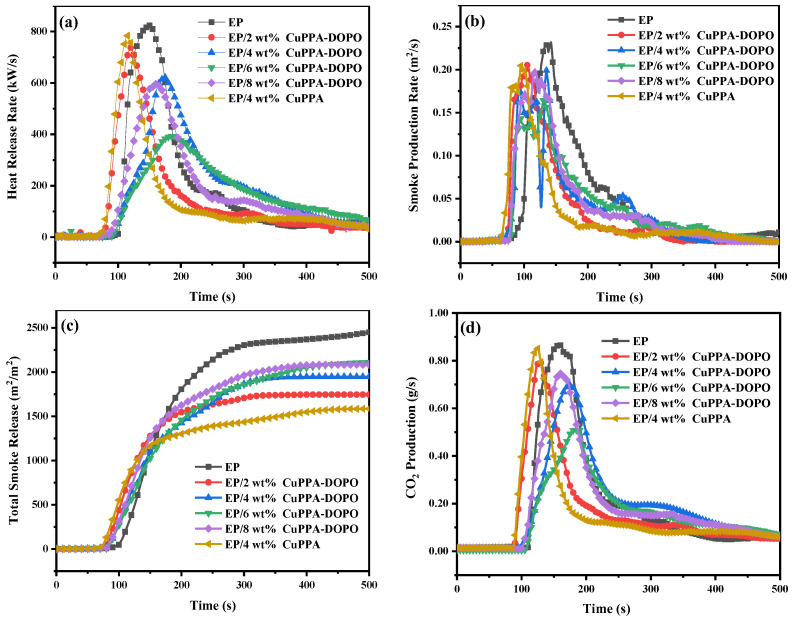
(**a**) HRR, (**b**) SPR, (**c**) TSR and (**d**) CO_2_P curves of pure EP and EP composites.

**Figure 6 molecules-28-00623-f006:**
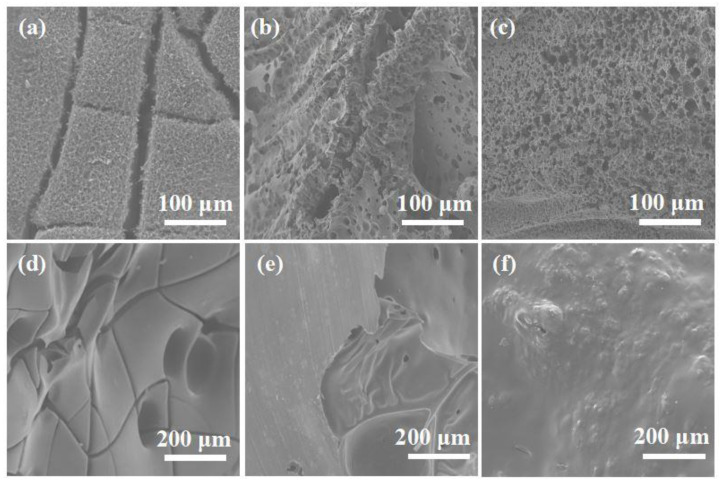
SEM images of external char layer: (**a**) EP; (**b**) EP/4 wt% CuPPA; (**c**) EP/4 wt% CuPPA-DOPO. SEM images of internal char layer: (**d**) EP; (**e**) EP/4 wt% CuPPA; (**f**) EP/4 wt% CuPPA-DOPO.

**Figure 7 molecules-28-00623-f007:**
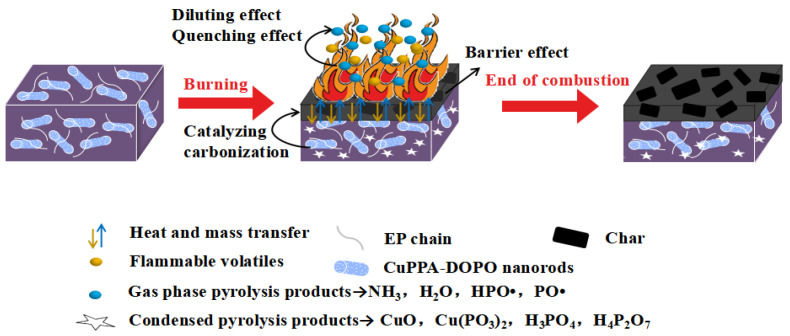
Flame-retardance mechanism diagram of EP/CuPPA-DOPO composites.

**Figure 8 molecules-28-00623-f008:**
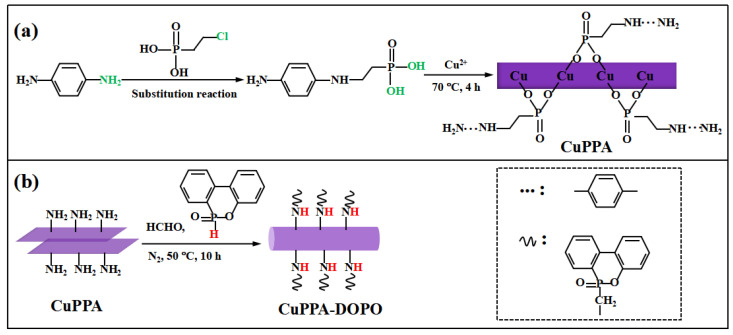
Synthetic route of (**a**) amino phenyl copper phosphate; (**b**) DOPO-amino phenyl copper phosphate.

**Figure 9 molecules-28-00623-f009:**
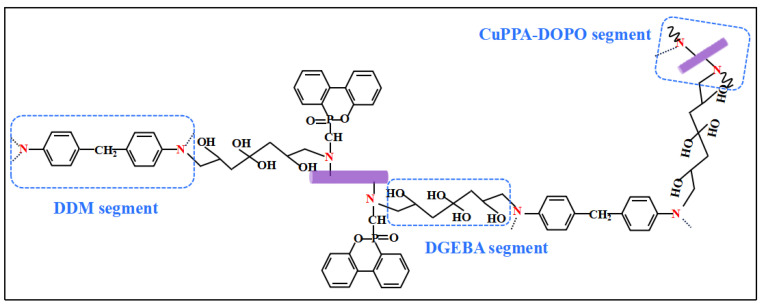
Schematic diagram of the curing reaction of EP composites.

**Table 1 molecules-28-00623-t001:** TGA data of pure EP and EP composites.

Samples	*T*_5%_ (°C)	*T*_max_ (°C)	Residues (wt%, 700 °C)
EP	360	382	14.9
EP/2 wt% CuPPA-DOPO	343	376	17.4
EP/4 wt% CuPPA-DOPO	333	373	23.0
EP/6 wt% CuPPA-DOPO	333	368	24.9
EP/8 wt% CuPPA-DOPO	342	372	26.1
EP/4 wt% CuPPA	350	376	19.8

**Table 2 molecules-28-00623-t002:** LOI data and UL-94 results of EP composites.

Samples	LOI (vol%)	UL-94			
		t_1_ (s)	t_2_ (s)	t_1_ + t_2_ (s)	Rating
EP	25.9 ± 0.2	-	-	>50	NR
EP/2 wt% CuPPA-DOPO	28.8 ± 0.2	37.0	6.3	43.3	V-1
EP/4 wt% CuPPA-DOPO	30.8 ± 0.3	20.5	7.0	27.5	V-1
EP/6 wt% CuPPA-DOPO	32.6 ± 0.3	13.6	5.0	18.6	V-1
EP/8 wt% CuPPA-DOPO	31.4 ± 0.3	14.5	9.0	23.5	V-1
EP/4 wt% CuPPA	28.2 ± 0.2	36.0	9.3	45.3	V-1

**Table 3 molecules-28-00623-t003:** CCT data of pure EP and EP composites.

Samples	PHRR (kW/m^2^)	PSPR (m^2^/s)	TSR (m^2^/m^2^)	CO_2_P (g/s)
EP	822	0.23	2438	0.87
EP/2 wt% CuPPA-DOPO	739	0.21	1746	0.79
EP/4 wt% CuPPA-DOPO	617	0.20	1942	0.70
EP/6 wt% CuPPA-DOPO	390	0.17	2112	0.51
EP/8 wt% CuPPA-DOPO	597	0.20	2085	0.75
EP/4 wt% CuPPA	784	0.21	1585	0.85

**Table 4 molecules-28-00623-t004:** Ingredients of EP composites.

Samples	Components			
	EP (wt%)	DDM (wt%)	CuPPA-DOPO (wt%)	CuPPA (wt%)
EP	80.0	20.0	0	0
EP/2 wt% CuPPA-DOPO	78.4	19.1	2	0
EP/4 wt% CuPPA-DOPO	76.8	17.1	4	0
EP/6 wt% CuPPA-DOPO	75.2	15.7	6	0
EP/8 wt% CuPPA-DOPO	73.6	14.2	8	0
EP/4 wt% CuPPA	76.8	19.2	0	4

## Data Availability

All data has been provided in this article.

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
