# Peer review of "Amino Phenyl Copper Phosphate-Bridged Reactive Phosphaphenanthrene to Intensify Fire Safety of Epoxy Resins"

_molecules, 2023, doi:10.3390/molecules28020623_

Round 1
Reviewer 1 Report
In this paper, CuPPA-DOPO was synthesized to prepare EP/CuPPA-DOPO composites. The results showed that CuPPA-DOPO could promote the premature decomposition of EP and increase the residual amount of the EP composites. The content is appropriate to be published on Molecules. However, there are some shortcomings in this manuscript, which should be revised before it is accepted.
1. In the part of 2.4 Preparation of EP/CuPPA-DOPO Composites, What is the ratio of resin and DDM curing agent?
2. In the part of 2.5 2.5. Measurements, in Limit oxygen index (LOI) and Vertical burning (UL-94) tests, how many times each sample is tested of samples?
3. In the parts of Thermal Properties of EP Composites, the action mechanism of CuPPA-DOPO and CuPPA is not clear and needs to be explained in detail.
4. In the CCT test, The TSR of EP/4 wt% CuPPA composites is 1585 m2/m2, which is 35% lower than pure EP and 18.4% lower than EP/4 wt% CuPPA-DOPO, indicating that CuPPA has better smoke suppression effect than CuPPA-DOPO. CUPPA has better smoke suppression effect than CUPPA-DOPO, there should be a reasonable explanation.
5. In the CCT test, the sentence of “But the CO2P of EP/4 wt% CuPPA composites is 0.85 g/s, which is slightly lower than pure EP and higher than EP/4 wt% CuPPA-DOPO. The decrease of CO2P of EP composites is attributed to the catalytic carbonization effect of the phosphorous compounds and copper oxides produced by the thermal cracking of CuPPA-DOPO [44,45].” The author should give a reasonable explanation for why there is such a result.
6. The article format should be consistent with the requirements of the Molecules.
Author Response
For Reviewer #1
We appreciate the reviewer #1 to give the comments and some revising suggestions about the manuscript, and have revised them taking into account the suggestions.
- 2.4. What is the ratio of resin and DDM curing agent for preparing EP/CuPPA-DOPO composite?
Replying: Thanks for your question. The mass ratio of resin and DDM curing agent is 4:1, which have been clarified in the revised manuscript.
2.4. Preparation of EP/CuPPA-DOPO Composites
Firstly, CuPPA-DOPO was dispersed in acetone and sonicated for 30 min. The preheated EP matrix was added in the mixture for 30 min of ultrasonic. Next, the mixture was stirred at 90 oC for 4 h until the acetone was completely removed. Then, DDM (mass ratio of EP matrix to DDM was 4:1) was added to the dispersion and stirred until DDM was completely dissolved. Then the mixture was quickly poured into the preheated silicone rubber mold and cured at 100 oC for 2 h. Finally, the splines were transferred to the drying oven and cured at 110 oC/2 h, 130 oC/2 h, 150 oC/2 h. The specific contents and curing reaction of EP composites are shown in Table 1 and Figure 2.
- At 2.5. Measurement: How many samples are tested for each sample in limiting oxygen index (LOI) and vertical combustion (UL-94) tests?
Replying: Thanks! To obtain even more precise results, 15 splines were burned and the mean values were obtained during the limiting oxygen index (LOI) test. In the vertical combustion experiment, 10 splines were burned and the mean values were obtained. The results have also been corrected in the revised manuscript.
Table 3. LOI data and UL-94 results of EP composites.
|
Samples |
LOI (vol%) |
UL-94 |
|||
|
t1 (s) |
t2 (s) |
t1+t2 (s) |
Rating |
||
|
EP |
25.9 ± 0.2 |
- |
- |
>50 |
NR |
|
EP/2 wt% CuPPA-DOPO |
28.8 ± 0.2 |
37.0 |
6.3 |
43.3 |
V-1 |
|
EP/4 wt% CuPPA-DOPO |
30.8 ± 0.3 |
20.5 |
7.0 |
27.5 |
V-1 |
|
EP/6 wt% CuPPA-DOPO |
32.6 ± 0.3 |
13.6 |
5.0 |
18.6 |
V-1 |
|
EP/8 wt% CuPPA-DOPO |
31.4 ± 0.3 |
14.5 |
9.0 |
23.5 |
V-1 |
|
EP/4 wt% CuPPA |
28.2 ± 0.2 |
36.0 |
9.3 |
45.3 |
V-1 |
- In the Thermal Properties of EP Composites section, the action mechanism of CuPPA-DOPO and CuPPA is not clear and needs to be explained in detail.
Replying: Thanks for your suggestion. According to your suggestion, we have given the thermal decomposition process of EP composites and the action mechanism of CuPPA-DOPO and CuPPA. In the thermal analysis, T50% and Tmax were introduced to illustrate the pre-decomposition effect of CuPPA-DOPO in EP matrix. The high temperature decomposition resistance of CuPPA-DOPO was verified by DTG curves. Then, the residual amount is used to illustrate the carbon forming capacity of the composites. Finally, the effects of each component on EP combustion are introduced.
3.2. Thermal Properties of EP Composites
The thermal stability of EP nanocomposites is an important evaluation index of fire safety. TGA and DTG curves and relevant data of EP nanocomposites are shown in Figure 5a, 5b and Table 2. The initial decomposition temperature (T5%) of EP is 360 oC, while EP/CuPPA-DOPO composites display the lower T5%. Among EP composites, T5% and Tmax (the temperature at maximum pyrolysis rate) of EP/6 wt% CuPPA-DOPO composites are decreased to 333 oC and 368 oC, respectively, which demonstrates that CuPPA-DOPO possesses a promoting influence of the decomposition for epoxy [31]. But the promoting effect of CuPPA is not obvious with T5% and Tmax to 350 oC and 376 oC. The decrease of maximum decomposition rate of EP/6 wt% CuPPA-DOPO composites also indicates that the presence of CuPPA-DOPO delays the degradation of the EP at a higher temperature [32]. Moreover, the incorporation of CuPPA-DOPO improves the char forming ability of the EP composites. The residues of EP composites are increased to 17.4%, 23.0%, 24.9% and 26.1%, respectively, when the amounts of CuPPA-DOPO are 2, 4, 6 and 8 wt%. While the residue of EP/4 wt% CuPPA composites at 700 oC is 19.8%, which is higher compared with that (14.9%) of pure EP, but is lower than EP/4 wt% CuPPA-DOPO. The outstanding char-forming ability of EP is attributed to the polyphosphoric substances and metallic oxide produced in the early decomposition of CuPPA-DOPO [33,34].
- In CCT test, the TSR of EP/4 wt% CuPPA composite is 1585 m2/m2, 35% lower than that of pure EP, and 18.4% lower than that of EP/4 wt% CuPPA-DOPO, indicating that CuPPA is superior to CuPPA-DOPO in smoke suppression. CUPPA has better smoke suppression effect than CUPPA-DOPO, which should be explained reasonably.
Replying: Thanks for your suggestion! In this work, CuPPA-DOPO was successfully synthesized, which was incorporated into EP matrix for preparing uniformly dispersed EP/CuPPA-DOPO composites. EP/CuPPA-DOPO composites exhibited good inhibition on heat release, which are attributed to that H2O, CO2 and NH2 released by the decomposition of CuPPA-DOPO can dilute the concentration of combustible volatile matter in the gas phase and absorb part of the combustion heat. The PO• produced by the thermal decomposition of CuPPA-DOPO produces a flame-retardant effect on the gas phase and suppresses the complete combustion of the gas phase products. However, the TSR value of EP/4 wt% CuPPA composites is 1585 m2/m2, which is 35% lower than pure EP and 18.4% lower than EP/4 wt% CuPPA-DOPO, because small phospho-oxygen molecules released after CuPPA-DOPO combustion become parts of the flue gas and enter the flame, thus increasing the TSR value and degenerating the smoke suppression performance.
- In the CCT test, "But the CO2P of EP/4 wt% CuPPA composites is 0.85 g/s, which is slightly lower than pure EP and higher than EP/4 wt% CuPPA-DOPO. The reduction of CO2P in EP composites is attributed to the catalytic carbonization of phosphorus compounds and copper oxide produced by CuPPA-DOPO thermal cracking [44,45]. The author should give a reasonable explanation for why this result occurs.
Replying: Thanks for your suggestion. The ability of EP/CuPPA to inhibit toxic gas release is not as good as that of EP/CuPPA-DOPO. The main reason is attributed to that CuPPA-DOPO plays flame retardant roles in the condensed phase and gas phase. The specific explanation was provided in the revised manuscript, which is as following
The CO2P of EP/4 wt% CuPPA composites is 0.85 g/s, slightly lower than pure EP and higher than EP/4 wt% CuPPA-DOPO. The main reason for the reduction of CO2P is that the combustion of CuPPA-DOPO can produce CuO and phosphorous flame retardant, catalyzing the formation of dense carbon layer to protect the surface of the composite materials. And the PO• by CuPPA-DOPO combustion exerts flame retardancy in the gas phase and inhibits the complete combustion of the gas phase products in the gas phase.
- The format of the article should meet the requirements of the Society.
Replying: Thanks for your suggestion. We have tried our best to adjust paper format to meet requirements.
- When multiple references are cited, e.g [6, 7] was modified into [6,7].
- The author contribution section is more standardized. The modified version is as follows. Conceptualization, Q.K., S.W. and J.Z.; methodology, C. and Z.L.; investigation, W.L. and Y.Z.; major experimental work and writing-original draft preparation, H.C.; writing-review and editing, Q.K., J.Z. and Y.Z.; resources, Q.K. and J.Z.; supervision, Y.Z. All authors have read and agreed to the published version of the manuscript.

Reviewer 2 Report
The manuscript under the title: “Amino Phenyl Copper Phosphate Bridged Reactive Phosphaphenanthrene to Intensify Fire Safety of Epoxy Resins” is in line with Polymers journal. This topic is relevant and will be of interest to the readers of the journal. It based on original research. This research has scientific novelty and practical significance. The article has a typical organization for research articles.
Before the publication it requires significant improvements, especially:
- The "Introduction" section: it has been proven that the effect of various modifying additives and fillers on the flammability reduction and physical and chemical properties of epoxy polymer composites is determined by many factors: ……. I think the related references should be cited corresponding to each aspect, e.g. (but not limited to these), which will undoubtedly improve the "Introduction" section:
- Polymers 2021, 13(15), 2421; https://doi.org/10.3390/polym13152421
- Polymers 2021, 13(19), 3332; https://doi.org/10.3390/polym13193332
- Inorg. Mater. Appl. Res. 2019, 10, 1135–1139, https://doi.org/10.1134/S2075113319050228
- Polymer Composites. 2020; 41: 2025–2035. https://doi.org/10.1002/pc.25517
- Section 2.1. It is necessary to add the physicochemical characteristics of components - give a table with the main physicochemical and technological properties of epoxy resin, hardener, DOPO and other components.
- Section 2.2. Why was the product not recrystallized to remove impurities?
- It is necessary to add data on the change in the viscosity of the epoxy composition with the introduction of fillers.
- Fig. 1 is confirmed by the FTIR data, it is necessary to confirm the proposed reactions in Fig. 2 with the FTIR data.
- For certain LOI values, a confidence interval or coefficient of variation must be specified.
- It is necessary to improve the quality of images for Fig.8.
- It is known that the introduction of dispersed fillers to an amount of more than 1 wt.% often leads to a decrease in the physical and mechanical characteristics of polymer composites. It would be nice to show how the filler you synthesized affects the strength of epoxy composites.
Author Response
For Reviewer #2
We appreciate the reviewer #2 to give the comments and some revising suggestions about the manuscript, and have revised them taking into account the suggestions.
- The "Introduction" section: it has been proven that the effect of various modifying additives and fillers on the flammability reduction and physical and chemical properties of epoxy polymer composites is determined by many factors: ……. I think the related references should be cited corresponding to each aspect, e.g. (but not limited to these), which will undoubtedly improve the "Introduction" section:
(1) Polymers 2021, 13(15), 2421; https://doi.org/10.3390/polym13152421
(2) Polymers 2021, 13(19), 3332; https://doi.org/10.3390/polym13193332
(3) Inorg. Mater. Appl. Res. 2019, 10, 1135–1139, https://doi.org/10.1134/S2075113319050228
(4) Polymer Composites. 2020; 41: 2025–2035. https://doi.org/10.1002/pc.25517
Replying: Thanks for your suggestion. According to your suggestion, we have read the relevant literatures and cited them in the introduction, which are following as:
- Zhou, S.; Tao, R.; Dai, P.; Luo, Z.Y.; He, M. Two-step fabrication of lignin-based flame retardant for enhancing the thermal and fire retardancy properties of epoxy resin composites. Polym. Compos. 2020, 41, 2025–2035.
- Mostovoy, A.S.; Nurtazina, A.S.; Kadykova, Y.A.; et al. Highly Efficient Plasticizers-Antipirenes for Epoxy Polymers. Inorg. Mater. Appl. Res. 2019, 10, 1135–1139.
- Bekeshev, A.; Mostovoy, A.; Kadykova, Y.; Akhmetova, M.; Tastanova, L.; Lopukhova, M. Development and Analysis of the Physicochemical and Mechanical Properties of Diorite-Reinforced Epoxy Composites. Polymers. 2021, 13, 2421.
- Bao, X.; Wu, F.; Wang, J. Thermal Degradation Behavior of Epoxy Resin Containing Modified Carbon Nanotubes. Polymers. 2021, 13, 3332.
- Section 2.1. It is necessary to add the physicochemical characteristics of components - give a table with the main physicochemical and technological properties of epoxy resin, hardener, DOPO and other components.
Replying: Thanks for your suggestion. According to your suggestion, we have provided the main physicochemical and technological properties of epoxy resin, hardener, DOPO and other components in Section 2.1. The specific modification is as follows.
2.1. Materials
Para-phenylene diamine (C6H8N2), potassium iodide (KI) and 2-chloroethyl phosphate (C2H6ClO3P) were supplied by Macklin Co., Ltd. 9,10-dihydro-9-oxygen-10-phospha-phenanthrene-10-oxide (DOPO) (phosphorus content, ≥14% by weigh), copper chloride dihydrate (CuCl2·2H2O), paraformaldehyde (HCHO), tetrahydrofuran (C4H8O), sodium hydroxide (NaOH) and diaminodiphenylmethane (DDM) (viscosity at 25 °C, 2.5-4.0 Pa∙s, amine value, 480 mg KOH g-1) were acquired from Sinopharm Chemical Reagent Co., Ltd. EP (NFEL128) (viscosity, 12-15 Pa∙s, epoxy equivalent, 184-190 g/mol) was bought from Nanya Electronic Materials (Kunshan) Co., Ltd.
- Section 2.2. Why was the product not recrystallized to remove impurities?
Replying: Thanks for your question! Based on our previous research experience, the impurities are soluble in deionized water and ethanol, which can be removed by multiple washing.
- It is necessary to add data on the change in the viscosity of the epoxy composition with the introduction of fillers.
Replying: Thanks for your suggestion. Generally speaking, the initial viscosity of the composites increases, the speed of viscosity change of the mixing system is accelerated, and the time required to reach the viscosity platform is shortened with the increase of the amount of nano filler. In this work, CuPPA-DOPO contains active N-H bond, which can chemically bind with epoxy on the molecular chain of epoxy resin, enhancing the interfacial binding between nanoparticles and epoxy matrix. According to the experimental phenomena, When the nanofiller was added to 8 wt%, EP composites became sticky but still poured and solidified into splines. If the addition amount continued to increase, the mobility of the epoxy resin composite became worse, and the preparation of the splines became difficult. I agree with you that the data on the change of epoxy composition viscosity on filler addition are important for understanding the interaction between the filler and epoxy. However, COVID-19 recently swept across China, and our laboratory has been completely closed, so we can’t test relevant data. In the future research, we will study this problem in depth.
5.Fig. 1 is confirmed by the FTIR data, it is necessary to confirm the proposed reactions in Fig. 2 with the FTIR data.
Replying: Thanks for your suggestion. The reaction in Figure 2 is a qualitative description based on theory and literature. CuPPA-DOPO contains secondary amine groups and can participate in the ring-opening reaction of epoxy resins. The hydrogen bond interaction between the intermediates -OH and -Ar-OH forms three-dimensional spatial network structure. References and reference mechanism diagrams are shown below: Wang J, Tang H, Yu X, Xu J, Pan Z, Zhou H. Reactive organophosphorus flame retardant for transparency, lowflammability, and mechanical reinforcement epoxy resin. J Appl Polym Sci. 2021;138:e50536. I agree with you that it is important to recognize this reaction that FTIR data were used to confirm the proposed reactions in Figure 2. However, COVID-19 recently swept across China, and our laboratory has been completely closed, so we can’t test relevant data. In the future research, we will study this problem in depth.
- For certain LOI values, a confidence interval or coefficient of variation must be specified.
Replying: Thanks for your suggestion. We have corrected the LOI values and provided confidence intervals in the revised manuscript, which is as following.
Table 3. LOI data and UL-94 results of EP composites.
|
Samples |
LOI (vol%) |
UL-94 |
|||
|
t1 (s) |
t2 (s) |
t1+t2 (s) |
Rating |
||
|
EP |
25.9 ± 0.2 |
- |
- |
>50 |
NR |
|
EP/2 wt% CuPPA-DOPO |
28.8 ± 0.2 |
37.0 |
6.3 |
43.3 |
V-1 |
|
EP/4 wt% CuPPA-DOPO |
30.8 ± 0.3 |
20.5 |
7.0 |
27.5 |
V-1 |
|
EP/6 wt% CuPPA-DOPO |
32.6 ± 0.3 |
13.6 |
5.0 |
18.6 |
V-1 |
|
EP/8 wt% CuPPA-DOPO |
31.4 ± 0.3 |
14.5 |
9.0 |
23.5 |
V-1 |
|
EP/4 wt% CuPPA |
28.2 ± 0.2 |
36.0 |
9.3 |
45.3 |
V-1 |
- It is necessary to improve the quality of images for Fig.8.
Replying: Thanks for your suggestion. According to your requirement, We have improved the quality of SEM images in Fig.8 in the revised manuscript, which is as following.
Figure 8. SEM images of external char layer: (a) EP; (b) EP/4 wt% CuPPA; (c) EP/4 wt% CuPPA-DOPO. SEM images of internal char layer: (d) EP; (e) EP/4 wt% CuPPA; (f) EP/4 wt% CuPPA-DOPO.
- It is known that the introduction of dispersed fillers to an amount of more than 1 wt% often leads to a decrease in the physical and mechanical characteristics of polymer composites. It would be nice to show how the filler you synthesized affects the strength of epoxy composites.
Replying: Thanks for your suggestion. In this work, our main aim was to investigate the effect of CuPPA-DOPO on the thermal properties and flame-retardant performances of EP composites. Usually, the addition of inorganic nanofillers leads to a decrease of mechanical properties of EP composites. However, adding a small amount of CuPPA-DOPO may have less effect on mechanical properties in this work, because both CuPPA and DOPO have organic properties and good compatibility with epoxy resin. We think it will have some influence on the mechanical properties of the material when the addition amount reaches 8 wt%, because the combustion performance of EP/8 wt% CuPPA-DOPO is worse than that of EP/6 wt% CuPPA-DOPO.

Reviewer 3 Report
Please correct three small editorial mistakes indocated in attached file

Author Response
We appreciate the reviewer #2 to give the comments and some revising suggestions about the manuscript, and have revised them taking into account the suggestions.
- About author, address and blank space
Replying: Thanks for your reminding! According to your suggestion, we have corrected the three minor errors in the revised manuscript, which is as following.

Round 2
Reviewer 2 Report
The authors considered most of the comments or adequately responded to the remarks contained in the review; therefore, the work may be approved for publication.